# The PAS-B Domain of BMAL1 Controls Proliferation, Cellular Energetics, and Inflammatory Response in Human Monocytic Cell Line THP-1

**DOI:** 10.3390/ijms26146737

**Published:** 2025-07-14

**Authors:** Yoko Gozu, Junichi Hosoi, Hiroaki Nagatomo, Kayako Ishimaru, Atsuhito Nakao

**Affiliations:** 1Mirai Technology Institute, Shiseido Co., Ltd., Kanagawa 220-0011, Japan; youko.gouzu@shiseido.com (Y.G.); jun1.hosoi.tomioka@gmail.com (J.H.); 2Center for Life Science Research, University of Yamanashi, Yamanashi 409-3898, Japan; hnagatomo@yamanashi.ac.jp; 3Department of Immunology, Faculty of Medicine, University of Yamanashi, Yamanashi 409-3898, Japan; kayako@yamanshi.ac.jp; 4Yamanashi GLIA Center, University of Yamanashi, Yamanashi 409-3898, Japan; 5Atopy Research Center, Juntendo University School of Medicine, Tokyo 113-8421, Japan

**Keywords:** BMAL1, monocytes, proliferation, cellular energetics, inflammation

## Abstract

*Brain muscle ARNT-like1* (*Bmal1*) is a transcriptional factor, consisting of basic helix–loop–helix (bHLH) and PER-ARNT-SIM (PAS) domains, that plays a central role in circadian clock activity. However, the precise roles of the BMAL1-PAS domain, a circadian rhythm-regulating structure, remain unexplored in monocytes. Here, we highlight the BMAL1-PAS domain as a key structure in monocyte pleiotropic functions by using human monocytic cell line THP-1. THP-1 cells lacking the BMAL1-PAS-B domain (THP-1#207) abrogated the circadian expression of core clock genes. THP-1#207 cells exhibited less proliferation, glycolysis and oxidative phosphorylation activity, and LPS-induced IL-1β production, but exhibited more production of LPS-induced IL-10 than THP-1 cells. A quantitative proteomics analysis revealed significant expression changes in ~10% metabolic enzymes in THP-1#207 cells compared to THP-1 cells, including reduction in a rate-limiting enzyme hexokinase2 (HK2) in the glycolytic pathway. Importantly, treatment of THP-1 with 2-deoxy-D-glucose (2-DG), an HK2 inhibitor, largely recapitulated the phenotypes of THP-1#207 cells. These findings suggest that the BMAL1-PAS-B domain is an important structure for the regulation of proliferation, cellular energetics, and inflammatory response in THP-1 cells, at least in part, via the control of glycolytic activity. Thus, the BMAL1-PAS-B domain may become a promising pharmacological target to control inflammation.

## 1. Introduction

Many living organisms adapt their physiological functions to a 24 h day–night rhythm (circadian rhythm) through an internal time keeping system called circadian clocks [1]. In mammals, the clocks consist of interlocking transcriptional–translational feedback loops (TTFLs) centered on the transcription factors BMAL1 and CLOCK [2,3]. BMAL1 heterodimerizes with CLOCK and the heterodimer binds to E-box motifs (CANNTG) throughout the genome, driving the expression of thousands of genes, including *Period (Per1-3)* and *Cryptochrome (Cry1,2)*. The PER and CRY proteins form oligomers and move to the nucleus, where they inhibit BMAL1/CLOCK activity. The core TTFLs are complemented by two additional TTFLs consisting of retinoic acid-related orphan receptors (*RORα*, *β*, *γ*) and *Reverb-a/b* and of *nuclear factor interleukin 3 regulated* (*NFIL3*) and *D-box binding PAR bZIP transcription factor* (*DBP*).

Among the clock genes, *Bmal1* has a prominent function within the circadian oscillators. *Bmal1* is the only non-redundant clock gene because single knock out mice for *Bmal1* become immediately arrhythmic after their release into constant dark conditions, whereas single knock out mice for *Clock*, *Cry1-3*, and *Per1,2* do not show such phenotypes [4]. BMAL1 has N-terminal basic helix–loop–helix (bHLH) DNA binding domain, tandem PER-ARNT-SIM (PAS) active domains (PAS-A and PAS-B), and C-terminal transcriptional activation domains (TADs), the structure of which is similar to CLOCK [5,6,7,8,9]. The PAS-domains of BMAL1 mediate heterodimerization with the PAS-domains of CLOCK, and mutations in the PAS-PAS interfaces between BMAL1 and CLOCK destabilize heterodimer formation, reduce transcriptional activity, and circadian periodicity, suggesting the essential role of the PAS domains in BMAL1/CLOCK complex activity [5,6,7,8,9].

Studies in the past ~10 years reveal that circadian clocks are not just a timekeeping system but play an important role in human health and diseases. In particular, recent studies have shown that one of the major targets of circadian clocks is immune function [10,11,12]. Accordingly, it is becoming clear that disruption of circadian clock activity by genetic or environmental factors such as shift work results in the disturbance of human immune response such as compromised resistance to infection. Therefore, detailed elucidation of the relationships between circadian clocks and immune cells is important to better understand immune function.

Monocytes are circulating innate immune cells generated from bone marrow precursors. Monocytes are rapidly recruited to sites of inflammation, where they differentiate into macrophages or dendritic cells and participate in the clearance of pathogens and dead cells, tissue healing, and initiation of adaptive immunity. It has been shown that circadian clocks intrinsic to monocytes (or macrophages) play a role in rhythmicity of monocyte (or macrophage) functions, such as metabolism, inflammatory mediator production, pathogen sensing, phagocytosis, and migration [13]. For instance, *Bmal1* regulates monocyte migration to the tissue in a circadian manner [14,15]. However, the roles of the BMAL1-PAS domain have not yet been explored in monocytes.

This study aimed to investigate the roles of the BMAL1-PAS domain in human monocyte functions. For this purpose, we used the human monocytic leukemia cell line THP-1. THP-1 cells maintain several monocytic nature [16] and have been utilized to investigate the functions of human monocytes as the most well-established monocyte line. We generated a BMAL1-PAS-B domain-deleted strain of THP-1 cells and used them to clarify the precise roles of the BMAL1-PAS-B domain in proliferation, cellular energetics, and inflammatory response in human monocytes.

## 2. Results

### 2.1. The BMAL1-PAS-B Domain Deletion Abrogates Circadian Expressions of Core Clock Genes in THP-1 Cells

We established THP-1 cells that selectively deleted the BMAL-PAS-B domain (termed THP-1#207) by double CRISPR/Cas9 genome editing (Figure 1a). DNA sequencing confirmed the deletion in the BMAL1-PAS-B domain in THP-1#207 cells (Appendix A). Interestingly, Western blot analysis using anti-human BMAL1 antibody targeting residues surrounding Gly552 in the C-terminal of human BMAL1 did not detect BMAL1 expression, suggesting that some co.nformational changes in BMAL1 might happen in THP-1#207 cells (Figure 1b). No off-target mutation was detected in THP-1#207 cells.

We then compared mRNA expressions of core clock genes *PER1*, *PER2*, *PER3*, *CRY 1*, *CRY2*, and *BMAL1* between unedited THP-1 cells (hereafter termed THP-1) and THP-1#207 cells at 6, 18, 30, and 42 h after synchronization with dexamethasone (100 nM) for 2 h. THP-1 cells exhibited circadian variations in *PER2*, *PER3*, *CRY1*, *CRY2*, and *BMAL1 mRNAs*, which was not observed in THP-1#207 cells (Figure 2). These findings suggested that THP-1 cells maintained circadian activity of core clock genes and the deletion of the BMAL1-PAS-B domain abrogated circadian expressions of core clock genes in THP-1 cells.

### 2.2. The BMAL1-PAS-B Domain Deletion Suppresses Proliferation in THP-1 Cells Associated with Low Activity of Glycolysis and Oxidative Phosphorylation

Interestingly, THP-1#207 cells showed significantly less proliferation than THP-1 cells as judged by direct cell counting and WST assay (Figure 3). Consistent with the proliferation phenotype, the basal mitochondrial respiratory activity (oxygen consumption rate: OCR) and glycolytic activity (extracellular acidification rate: ECAR) of THP-1#207 cells were significantly lower than those of THP-1 cells (Figure 4a). Low levels of basal respiration, ATP-linked respiration, and maximum respiration were also observed (Figure 4b). These findings suggested that the deletion of the BMAL1-PAS-B domain suppressed proliferation in THP-1 cells associated with low cellular energetics.

### 2.3. The BMAL1-PAS-B Domain Deletion Suppresses LPS-Induced IL-1b, but Increases IL-10 Production in THP-1 Cells

Since cellular metabolic pathways, in particular the glycolytic activity, controls monocytes/macrophages activation in infectious context [17], we compared LPS-stimulated IL-1β and IL-10 production between THP-1#207 cells and THP-1 cells. LPS-stimulated IL-1β production decreased whereas LPS-stimulated IL-10 production increased in THP-1#207 cells compared to THP-1 cells (Figure 5).

These findings suggested that the deletion of the BMAL1-PAS-B domain suppressed LPS-induced IL-1β, but increased IL-10 production in THP-1 cells.

### 2.4. Inhibition of the Glycolytic Activity in THP-1 Cells Largely Recapitulates Phenotypes of THP-1#207 Cells

To gain a mechanistic insight into how the BMAL1-PAS-B domain deletion affects proliferation, cellular energetics, and LPS-induced cytokine production in THP-1 cells, we performed an in vitro large-scale targeted proteomics assay termed “iMPAQT” which enables the absolute quantification of metabolic enzymes and delineates the metabolic landscape [18]. The iMPACT analysis detected 278 enzymes, of which 32 showed a difference of 1.5-fold or more in THP-1#207 cells compared to THP-1 cells (Appendix A, Figure 6a). In particular, the expression levels of the two rate-limiting enzymes in the glycolytic pathway, hexokinase 2 (HK2), phosphofructokinase (PFKP), and lactate dehydrogenase B (LDHB), were significantly lower in THP-1#207 cells than in THP-1 cells (Figure 6b). Additionally, glutamate dehydrogenase 1 (GLUD1), an enzyme in TCA cycle, and branched chain amino acid transaminase 1 (BCAT1), an enzyme in amino acid metabolism, were also significantly lower in THP-1#207 cells than in THP-1 cells (Figure 6b). These findings suggested that the BMAL1-PAS-B domain deletion decreased protein expression levels of ~10% of metabolic enzymes and, in particular, reduced the expression of key enzymes HK2, PFKP, and LDHB in the glycolytic pathway in THP-1 cells (Figure 6c).

Since the BMAL1-PAS-B domain deletion reduced HK2, PFKP, and LDHB in glycolytic pathway in THP-1 cells, we asked whether reduction in the enzymes in the glycolytic pathway (and possible reduction in the glycolytic activity) by the Bmal1-PAS-B-domain deletion could underlie the phenotypes observed in THP-1 #207 cells. For this purpose, we examined the effects of 2-deoxy-D-glucose (2-DG), an HK2 inhibitor, on proliferation, cellular energetics, and LPS-induced cytokine production in THP-1 cells. 2-DG suppressed proliferation, glycolysis, and oxidative phosphorylation activities and LPS-induced IL-1β, but not IL-10, production in THP-1 cells (Figure 7). These findings suggested that glycolytic activity played an important role for the regulation of proliferation, cellular energetics, and a part of inflammatory response (LPS-induced IL-1β production) in THP-1 cells. Thus, the treatment of THP-1 with 2-DG largely recapitulated the phenotypes observed in THP-1#207 cells.

## 3. Discussion

This study showed that THP-1 cells lacking the BMAL1-PAS-B domain (THP-1#207) exhibited less proliferation, glycolytic, and oxidative phosphorylation activity, and LPS-induced IL-1β production than THP-1 cells. Quantitative proteomics analysis showed reduction in HK2, a rate-limiting enzyme in the glycolytic pathway, expression levels in THP-1#207 cells. Importantly, treatment of THP-1 cells with an inhibitor of HK2 largely recapitulated the in vitro phenotypes observed in THP-1#207 cells. These findings suggest that the BMAL1-PAS-B domain is an important structure that controls proliferation, cellular energetics, and inflammatory response in THP-1 cells, at least in part, via the regulation of the glycolytic activity.

The current results suggest that the BMAL1-PAS-B domain positively controls glycolytic activity, which underlies proliferation and LPS-induced inflammation in THP-1 cells. Consistent with the current findings, Harfmann et al. showed that Bmal1 deficiency decreased glycolytic flux in mouse muscle (myocytes) [19]. In addition, Wang et al. showed that BMAL1 positively controls glycolytic activity in trastuzumab-resistant HER2-positive gastric cancer cells [20]. Importantly, both studies showed that the BMAL1 regulation of HK2 expression is a key to promoting glycolytic activity, which is also consistent with the current study. Taken together, the BMAL-1-PAS-B domain may be an important structure for BMAL1-control of glycolytic activity, at least in part, by regulating HK2 expression.

However, there are several opposite studies suggesting that Bmal1 negatively controls glycolytic activity. For instance, *Bmal1*-deficient macrophages displayed an increased glycolytic activity in mice [21]. In addition, BMAL1 overexpression restrained glycolytic activity to repress M1-like macrophage polarization in mice [22]. The discrepancy may be explained by the usages of different cell types (myocytes, gastric cancer cells, THP-1 cells, and primary mouse macrophages), different species (human vs. mouse), and different experimental conditions. Therefore, the precise roles of BMAL1 or the BMAL1-PAS-B domain in the glycolytic activity on pleiotropic biological context remain to be determined in future studies.

THP-1#207 cells also exhibited impaired oxidative phosphorylation activity as well as glycolytic activity. This might be related to reduction in GLUD1, an enzyme in TCA cycle (Figure 6), and also be associated with less proliferation observed in THP-1#207 cells. Consistently, several studies suggest that BMAL1 controls the mitochondrial bioenergetics by regulating transcriptional factors such as PGC-1α [23,24,25]. We would like to perform more detailed analysis about mitochondrial bioenergetics such as function of mitochondrial electric transport chain in THP#207 cells in our future studies.

Treatment of 2-DG did not affect LPS-induced IL-10 production in THP-1 cells, suggesting that the BMAL1-PAS-B domain deletion may enhance LPS-induced IL-10 production independently of suppression of the glycolytic activity. Recent studies suggest that LPS-induced IL-10 production depends on redox states, in particular, ROS production in macrophages [26,27]. Thus, it will also be interesting and important to investigate mitochondrial function in detail such as ROS production in THP-1#207 cells.

There are several limitations of this study. First, although THP-1 cells maintain monocyte nature and are widely utilized to investigate the functions of human monocytes [16], the findings obtained from THP-1 cells derived from a patient with monocytic leukemia could not reflect physiological functions of monocytes. As the second limitation, in our current study, except for the experiments observing the dynamics of clock genes (Figure 2), we did not apply synchronization stimuli to THP-1 cells. Therefore, the effects observed in these experiments primarily reflect the consequences of BMAL1-PAS-B domain deficiency, and the relationship between these results and the circadian clock remains unclear. Third, we should compare THP-1 cells, THP-1#207 cells, and THP-1 cells deleting entire *BMAL1* gene to clarify the specific roles of PAS-B domain in monocyte function. This issue will be addressed in our future studies.

In summary, we suggest that the PAS-B domain is an important structure of BMAL1 that controls proliferation, cellular energetics, and inflammatory response in THP-1 cells, largely by regulating the glycolytic activity. Thus, we assume that the BMAL1-PAS-B domain plays a critical role in monocytes/macrophages function. Consistent with current findings, Pu et al. most recently reported that pharmacological targeting of the BMAL-1-PAS-B-domain impacts inflammatory and phagocytic function in macrophages [7]. Further, given consideration that the circadian clock plays a crucial role in a variety of human diseases including inflammation, metabolism, and cancer [12,28,29] and BMAL1 is a core circadian protein [4], the BMAL1-PAS-domain may become a promising pharmacological target to control not only inflammation but also many circadian-related human disorders.

## 4. Materials and Methods

### 4.1. Cell Culture

Human monocytic leukemia cell line THP-1 cells (ATCC; TIB-202) [16] or THP-1#207 cells (please see below) were cultured in RPMI1640 medium (Nacalai Tesque, Kyoto, Japan) supplemented with 10% fetal bovine serum (FBS, Nichirei bioscience, Tokyo, Japan), 1% P/S (Fujifilm Life Science, Tokyo, Japan), MEM essential amino acids (Fujifilm Life Science), MEM non-essential amino acids (Nacalai Tesque), sodium pyruvate (Nacalai Tesque), and L-glutamate (Nacalai Tesque), at 37 °C under 5% CO_2_.

### 4.2. Establishment of BMAL1-PAS-B Domain-Deleted Cell Line THP-1#207 Cells

The double CRIPR/Cas9 technique was used to establish BMAL1-PAS-B domain deletion. sgRNAs were designed in the PAS-B domain exon region and placed tail to tail on complementary strands within 20 bases, including PAM regions, by using CRIPR direct software [30]. Synthetic sgRNAs and Cas9 nuclease (Thermo Fisher Scientific, Waltham, MA, USA) were transfected by lipofection (Lipofectamine CRISPR MAX reagent, Thermo Fisher Scientific) with puromycin resistance synthetic DNA (Takara Bio, Shiga, Japan). Two days after transfection, 1 µg/mL puromycin was added to the medium, then living cells were selected after an additional three days. Mutation of the target region was confirmed by DNA sequencing of PCR-amplified DNA with target region-specific primers.

### 4.3. Quantitative Real-Time PCR (qPCR)

Total RNA was prepared by using RNeasy mini kit (Qiagen, Hilden, German) and was used for cDNA synthesis (PrimeScript 1st strand cDNA Synthesis Kit, Takara Bio). Relative gene expression levels were analyzed by real-time PCR (AriaMx, Agilent Technologies, Santa Clara, CA, USA) with TB Green (TB Green Ex TaqII, Takara Bio). The expression of *GAPDH* was used as an internal control for normalization.

### 4.4. Western Blot

Protein expression of the target region was confirmed by Western blotting. Protein was extracted with RIPA buffer, separated by SDS-PAGE (5–15% Bullet Page Precast Gel, Nacalai Tesque), and transferred to PVDF membranes (Thermo Fisher Scientific). Membranes were incubated with rabbit anti-human BMAL1 monoclonal IgG antibody (Cell Signaling Technology, Danvers, MA, USA, #14020) targeting to residues surrounding Gly552 in the C-terminal of human BMAL1 protein. Then reacted with HRP-conjugated secondary antibody (Thermo Fisher Scientific). ECL signal was detected with ECL detection reagent (Novex AP Chemiluminescent Detection Kit, ThermoFisher Scientific) and a chemiluminescence imaging system (Fusion FX, Vilber, Lourmat, France). b-actin (Santa Cruz Biotechnology, Dallas, TX, USA, #sc47778) was used as the control.

### 4.5. Synchronization

For qPCR analysis of core clock genes, THP-1 cells and THP-1#207 cells were synchronized using 100 nM dexamethasone (DEX) (Sigma-Aldrich, St. Louis, MO, USA; D4902) for 2 h, as previously described [31]. After medium change, the cells were cultured for 6, 18, 30, and 42 h and harvested for the analysis at each time point.

### 4.6. Cell Proliferation

THP-1 cells or THP-1#207 cells (3 × 10^3^ cells/well) in a 96-well plate and were cultured for 9 days, and the number of the cells was subsequently counted by a hemacytometer at day 1, day4, day7, and day10. In addition, cell proliferation was monitored using the Cell Counting Kit-8 (WST assay) (Dojindo Laboratory, Kumamoto, Japan). For 2-Deoxy-D-glucose (2-DG) (Fujifilm Life Science) experiments, THP-1 cells and THP-1#207 cells were cultured in the presence or absence of 0.3 mM or 3 mM 2-DG for the indicated time.

### 4.7. Extracellular Flux Analysis

A Seahorse XFe24 analyzer (Agilent Technologies) and XF Mitostress Test Kit (Agilent Technologies) were used for the analysis of cell energy metabolism. Oxygen consumption rate (OCR, pmol/min) and extracellular acidification rate (ECAR, mpH/min), which were normalized to living cell number determined by staining with Hoechst33342 (Thermo Fisher Scientific, #H1399), were measured as parameters of cell energy activity. A total of 100,000 cells/well were seeded into a Seahorse XF 24-well plate precoated with CELL-TAK (Corning, Bedford, MA, USA). Seahorse XF DMEM medium (pH 7.4) supplemented with 10 mM glucose, 1 mM pyruvate, 2 mM L-glutamate was used as the assay medium. Basal respiration rate was measured for the first 3 min, then 100 µM oligomycin, 100 µM FCCP, and 50 µM rotenone/antimycin (ROT/AA) were added sequentially. The difference between the OCR value after oligomycin addition and the basal value was taken as ATP respiration activity, and the difference between the OCR value after FCCP addition and that after ROT/AA addition was taken as the maximum respiration activity. For 2-DGexperiments, THP-1 cells and THP-1#207 cells were cultured in the presence or absence of 0.3 mM or 3 mM 2-DG for the indicated time.

### 4.8. Enzyme-Linked Immunosorbent Assay (ELISA)

THP-1 cells and THP-1#207 cells were stimulated with 100 ng/mL of lipopolysaccharide (LPS, Sigma-Aldrich) for 24 h and the concentrations of IL-1β and IL-10 in supernatants were determined by human IL-1β and IL-10 ELISA kit (R&D systems, Minneapolis, MN, USA). For 2-DG experiments, THP-1 cells or THP-1#207 cells were treated with 0.3 mM or 3 mM 2-DG for 24 h followed by stimulation with 100 ng/mL LPS for 24 h. Then, the culture supernatants were subjected to cytokine analysis.

### 4.9. iMPAQT Proteomics

THP-1 cells and THP-1#207 cells (2 × 10^6^ cells) were subjected to the next-generation proteomics, in vitro proteome-assisted Multiple Reaction Monitoring (MRM) for Protein Absolute QuanTification (iMPAQT) (KPSL, Fukuoka, Japan) as previously described [18].

### 4.10. Statistical Analysis

Data are presented as individual values or mean ± SD, and GraphPad PRISM v9.5.1software (GraphPad Software, La Jolla, CA, USA) was used for statistical analysis. Student’s *t*-test (two-tailed) was used for comparison of two values, and analysis of variance (ANOVA, with the Bonferroni test) was performed for multi-value analysis. Significance values are shown in each figure.

## Figures and Tables

**Figure 1 ijms-26-06737-f001:**
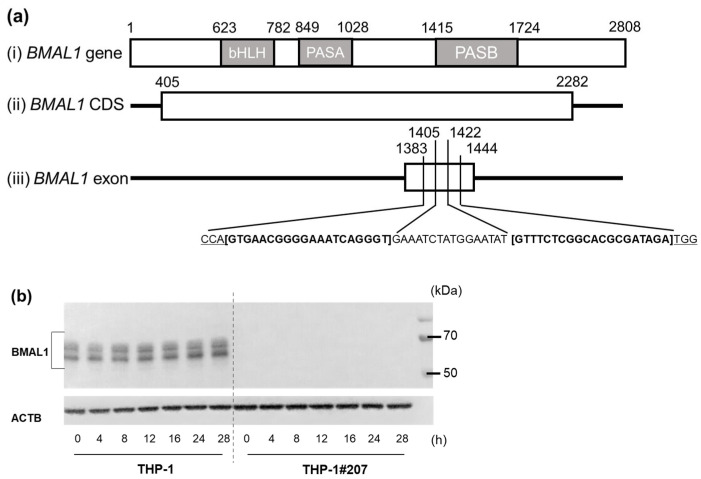
Generation of BMAL1-PAS-B domain-deleted THP-1 cells (THP-1#207 cells). (**a**) Construction of BMAL1 gene (i), coding DNA sequence (CDS) of BMAL1 (ii), and exon targeted by CRISPR/Cas9. The protospacer adjacent motif (PAM) sequence and sgRNA sequence are indicated (iii). (**b**) Western blot analysis with anti-BMAL1 antibody.

**Figure 2 ijms-26-06737-f002:**
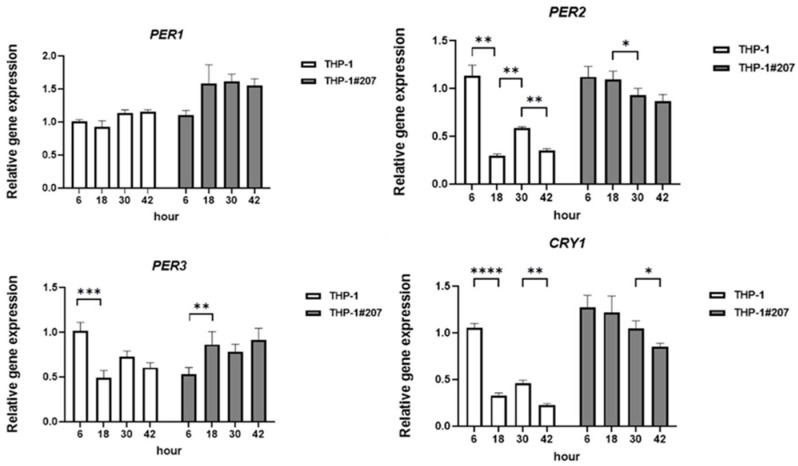
The BMAL1-PAS-B domain deletion abrogates circadian expression of core clock genes in THP-1 cells. Time course of PER1-3, CRY1-2, and BMAL1 mRNA expression in THP-1 cells or THP-1#207 cells after synchronization with dexamethasone (DEX). Statistical analysis was performed using two-way ANOVA (with Bonferroni test). Data are presented as mean + SD, * *p* < 0.05, ** *p* < 0.01, *** *p* < 0.001, **** *p* < 0.0001.

**Figure 3 ijms-26-06737-f003:**
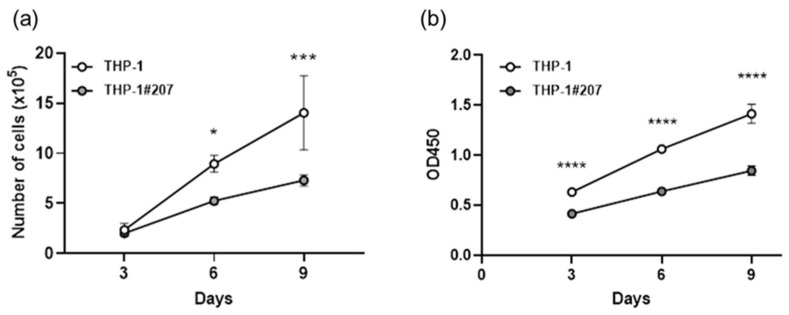
The BMAL1-PAS-B domain deletion suppresses proliferation in THP-1 cells. THP-1 cells or THP-1#207 cells (1 × 10^5^ cells/mL) were cultured for 3, 6, 9 days and the cell numbers were counted by direct cell counting (**a**) or Water Soluble Tetrazolium salts (WST) assay (**b**). Statistical analysis was performed using two-way ANOVA (with Bonferroni test). Data are presented as mean ± SD, * *p* < 0.05, *,* *** *p* < 0.001, **** *p* < 0.0001.

**Figure 4 ijms-26-06737-f004:**
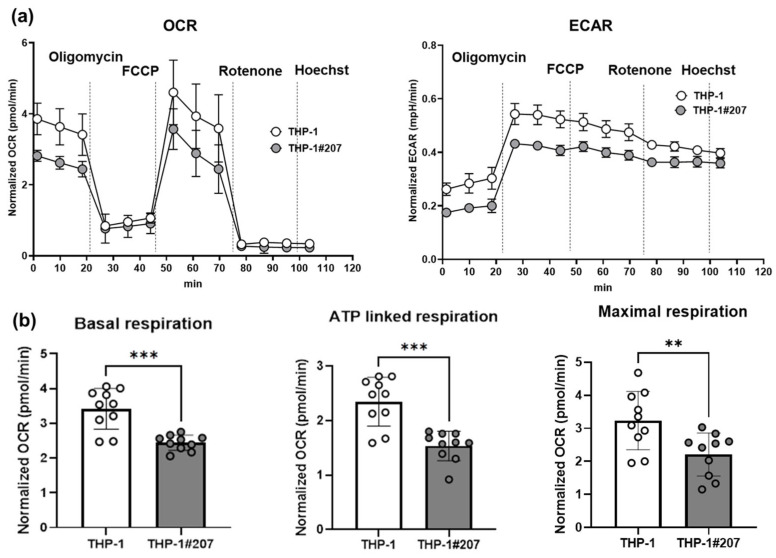
The BMAL1-PAS-B domain deletion suppresses glycolysis and oxidative phosphorylation in THP-1. (**a**) Oxygen Consumption Rate (OCR) and Extracellular Acidification Rate (ECAR) were measured following the injection of the mitochondrial stress test compounds oligomycin, p-Trifluoromethoxy phenylhydrazine carbonyl cyanide (FCCP), and rotenone–antimycin A. After measurement, the number of living cells was counted by Hoechst staining for normalization. N = 10, mean ± SD. (**b**) Basal respiration, ATP-linked respiration, and maximal respiration calculated from (**a**). N = 10, Student’s *t*-test, ** *p* < 0.01, *** *p* < 0.001.

**Figure 5 ijms-26-06737-f005:**
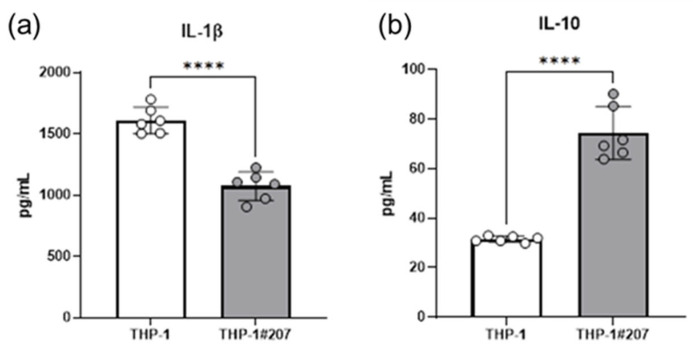
The BMAL1-PAS-B domain deletion affects LPS-induced IL-1β and IL-10 production in THP-1 cells. THP-1 cells or THP-1#207 cells (1 × 10^5^ cells/mL) were stimulated with LPS (100 ng/mL) and cultured for 24 h. Then, IL1β (**a**) and IL-10 (**b**) in the culture supernatants were measured by ELISA. Statistical analysis was performed using Student’s *t*-test. **** *p* < 0.0001.

**Figure 6 ijms-26-06737-f006:**
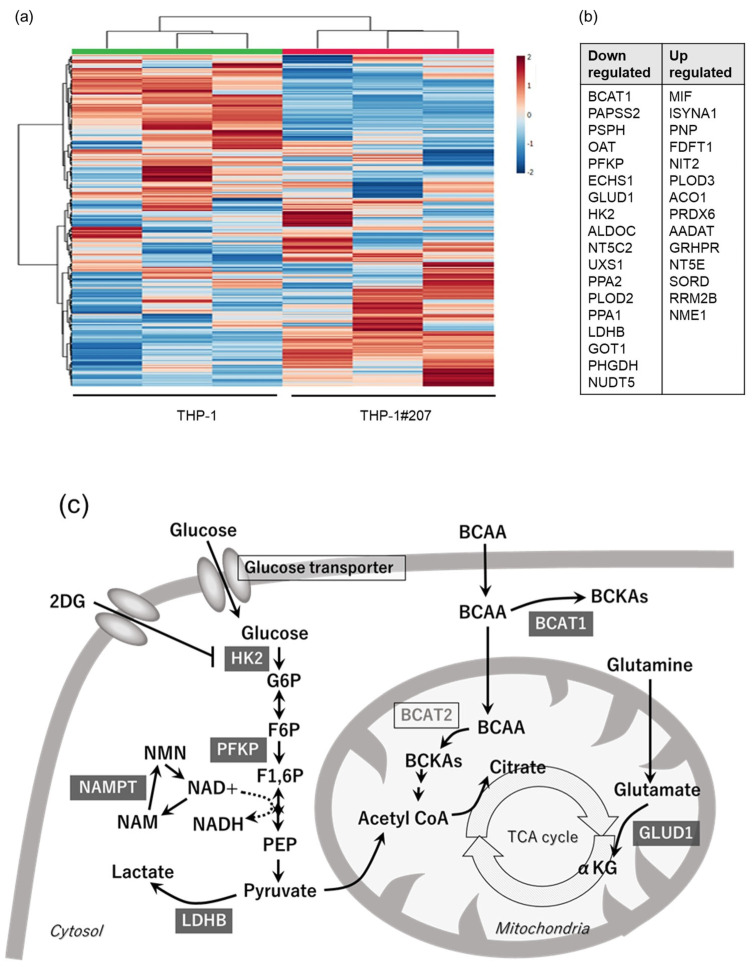
The BMAL1-PASB domain deletion affects the expression levels of metabolic enzymes. (**a**) The heat map of the proteomics analysis. Each column represents three different cell lines in THP-1 and THP-1#207. (**b**) A total of 32 metabolic enzymes showed a variation of 1.5-fold or more. A total of 18 enzymes were downregulated, and 14 enzymes were upregulated in THP-1#207. (**c**) Cellular energy metabolism cascade. Downregulated enzymes in THP-1#207 were shown with gray boxes.

**Figure 7 ijms-26-06737-f007:**
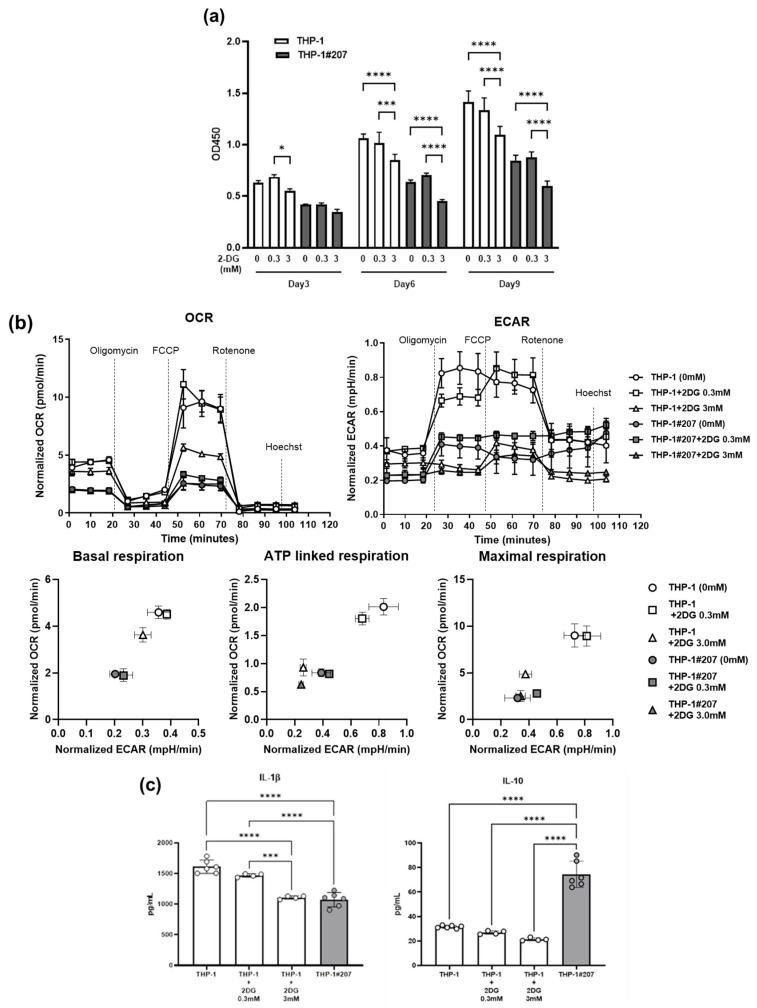
Inhibition of HK2 suppresses proliferation, cellular energetics, and LPS-induced IL-1βproduction in THP-1 cells, THP-1 cells or THP-1#207 cells (1 × 10^5^ cells/mL) were cultured for 24 h in the presence or absence of 2-deoxy-D-glucose (2-DG) (0.3 mM and 3 mM). (**a**) Cells were cultured for 3, 6, 9 days and cell numbers were evaluated by Water Soluble Tetrazolium salts (WST) assay. (**b**) Oxygen Consumption Rate (OCR) and Extracellular Acidification Rate (ECAR) were measured after with or without 2-DG treatment. (**c**) Cells were stimulated with LPS (100 ng/mL) and cultured presence or absence of 2-DG (0.3 mM and 3 mM). Then, IL-1βand IL-10 in the culture supernatants were measured by ELISA. Statistical analysis was performed using two-way ANOVA (with Bonferroni test). * *p* < 0.05, *** *p* < 0.001, **** *p* < 0.0001.

## Data Availability

The original contributions presented in this study are included in the article/Appendix A. Further inquiries can be directed to the corresponding author.

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
