# Peer review of "The PAS-B Domain of BMAL1 Controls Proliferation, Cellular Energetics, and Inflammatory Response in Human Monocytic Cell Line THP-1"

_ijms, 2025, doi:10.3390/ijms26146737_

Round 1
Reviewer 1 Report
Comments and Suggestions for Authors
The article "The PAS-B domain of BMAL1 controls proliferation, cellular energetics, and inflammatory response in human monocytic cell line THP-1" by Yoko Gozu et all it is very interesting.
The purpose of this study was to study the role of the BMAL1-PAS domain in human monocyte functions using the human monocytic leukemia THP-1 cell line.
However, several points have to be addressed:
- Abstract does not contain information - for what application? For example, it would be possible to add that the BMAL1-PAS-B domain may become a promising pharmacological target to control immunity.
- In the discussion, I recommend expanding the list of references and describing the possibilities of applying the results.
- I hope that the authors will compare THP-1 cells, THP-1#207 cells, and THP-1 cells with the deleted BMAL1 gene to clarify the specific role of the PAS-B domain in monocyte function in their future studies.
Author Response
Reviewer 1:
The article "The PAS-B domain of BMAL1 controls proliferation, cellular energetics, and inflammatory response in human monocytic cell line THP-1" by Yoko Gozu et al, it is very interesting.
Thank you very much for taking the time to review our manuscript. We appreciate your valuable feedback and suggestions for improvement. We have taken diligent consideration of your comments and have undertaken substantial revisions to adequately address the raised concerns. Please refer to the responses below for our detailed discussion and revision regarding your concerns.
The purpose of this study was to study the role of the BMAL1-PAS domain in human monocyte functions using the human monocytic leukemia THP-1 cell line.
However, several points have to be addressed:
- Abstract does not contain information - for what application? For example, it would be possible to add that the BMAL1-PAS-B domain may become a promising pharmacological target to control immunity.
Thank you for your comment. In the revised manuscript, we have added the statements in the Abstract that "the BMAL1-PAS-B domain may become a promising pharmacological target to control inflammation".
- In the discussion, I recommend expanding the list of references and describing the possibilities of applying the results.
Thank you for your comment to make our paper more attractive. In the revised manuscript, we have added the statements in the final paragraph of Discussion as follows with several references on the roles of the circadian clock (Bmal1) and human disease.
" In summary, we suggest that the PAS-B domain is an important structure of BMAL1 that controls proliferation, cellular energetics, and inflammatory response in THP-1 cells, largely by regulating the glycolytic activity. Thus, we assume that the BMAL1-PAS-B domain plays a critical role in monocytes/macrophages function. Consistent with current findings, Pu et al. most recently reported that pharmacological targeting of the BMAL-1-PAS-B-domain impacts inflammatory and phagocytic function in macrophages [7]. Further, given consideration that the circadian clock plays a crucial role in a variety of human diseases including inflammation, metabolism, and cancer [12, 28, 29] and BMAL1 is a core circadian protein [4], the BMAL1-PAS-domain may become a promising pharmacological target to control not only inflammation but also many circadian-related human disorders."
- I hope that the authors will compare THP-1 cells, THP-1#207 cells, and THP-1 cells with the deleted BMAL1 gene to clarify the specific role of the PAS-B domain in monocyte function in their future studies.
Thank you for the fundamental question. As we stated in the Discussion of the original (and revised) manuscript (please see below), we acknowledge the limitation of our paper. We would like to clarify the specific roles of the BMAL1-PAS-B domain in THP-1 cells in our future study.
"Third, we should compare THP-1 cells, THP-1#207 cells, and THP-1 cells deleting entire BMAL1 gene to clarify the specific roles of PAS-B domain in monocyte function. This issue will be addressed in our future studies."
Reviewer 2 Report
Comments and Suggestions for Authors
This detailed, quite technical and specific article, intended more for specialists in the field than for the general public, deserves to be published, but before that it needs some improvements:
Line 286-when you discuss the synchronization of cells with dexamethasone; how did you prevent nonspecific effects in the long term? because this aspect can even influence the inflammatory response.
Line 377-although it is a reference title in this field, it is an old source and it is recommended to replace it with newer research in the field.
Lines 389-396- the numbering of the sources is doubled.
General Bibliography: there are other current bibliographic sources related to the subject that are worth mentioning.
Author Response
Reviewer 2:
This detailed, quite technical and specific article, intended more for specialists in the field than for the general public, deserves to be published, but before that it needs some improvements:
Thank you for the precious time carefully reviewing our manuscript and providing valuable comments. We appreciate your positive comments and insightful suggestions to improve our manuscript. We have carefully considered your comments and have made revisions to address the concerns raised. Please refer to our responses below for details.
Line 286-when you discuss the synchronization of cells with dexamethasone; how did you prevent nonspecific effects in the long term? because this aspect can even influence the inflammatory response.
Thank you for your fundamental question. Actually, except for the experiments of Figure 2, we did not apply synchronization stimuli to THP-1 cells. Thus, we assume that dexamethasone would not affect inflammatory response in THP-1 cells. However, the absence of synchronization stimuli in the cells may introduce an additional limitation (please see below). Thus, we have added the statements in the Discussion of the revised manuscript as follows;
"As the second limitation, in our current study, except for the experiments observing the dynamics of clock genes (Figure 2), we did not apply synchronization stimuli to THP-1 cells. Therefore, the effects observed in these experiments primarily reflect the consequences of BMAL1 PAS-B domain deficiency, and the relationship between these results and the circadian clock remains unclear."
Line 377-although it is a reference title in this field, it is an old source and it is recommended to replace it with newer research in the field.
Thank you for your precious suggestion.
We have added new references in the revised manuscript as follows;
Ruan W, Li T, Bang IH, Lee J, Deng W, Ma X, Luo C, Du F, Yoo SH, Kim B, Li J, Yuan X, Figarella K, An YA, Wang YY, Liang Y, DeBerge M, Zhang D, Zhou Z, Wang Y, Gorham JM, Seidman JG, Seidman CE, Aranki SF, Nair R, Li L, Narula J, Zhao Z, Gorfe AA, Muehlschlegel JD, Tsai KL, Eltzschig HK.BMAL1-HIF2A heterodimer modulates circadian variations of myocardial injury. Nature. 2025;641:1017-1028.
Wu D, Rastinejad F. Structural characterization of mammalian bHLH-PAS transcription factors. Curr Opin Struct Biol. 2017;43:1-9.
Lines 389-396- the numbering of the sources is doubled.
We are sorry for our careless mistake. We have corrected the errors in the revised manuscript.
General Bibliography: there are other current bibliographic sources related to the subject that are worth mentioning.
Thank you very much for your comments. In the revised manuscript, we have added 5 new references related to this subject and published recently in the revised manuscript as follows;
(New references)
Ruan W, Li T, Bang IH, Lee J, Deng W, Ma X, Luo C, Du F, Yoo SH, Kim B, Li J, Yuan X, Figarella K, An YA, Wang YY, Liang Y, DeBerge M, Zhang D, Zhou Z, Wang Y, Gorham JM, Seidman JG, Seidman CE, Aranki SF, Nair R, Li L, Narula J, Zhao Z, Gorfe AA, Muehlschlegel JD, Tsai KL, Eltzschig HK. BMAL1-HIF2A heterodimer modulates circadian variations of myocardial injury. Nature. 2025;641:1017-1028.
Schrader LA, Ronnekleiv-Kelly SM, Hogenesch JB, Bradfield CA, Malecki KM. Circadian disruption, clock genes, and metabolic health. J Clin Invest. 2024;134:e170998.
Fortin BM, Mahieu AL, Fellows RC, Kang Y, Lewis AN, Ead AS, Lamia KA, Cao Y, Pannunzio NR, Masri S. The diverse roles of the circadian clock in cancer. Nat Cancer. 2025;6:753-767.
Wu D, Rastinejad F. Structural characterization of mammalian bHLH-PAS transcription factors. Curr Opin Struct Biol. 2017;43:1-9.
Balsalobre A, Marchacci L, Schibler U. Multiple signaling pathways elicit circadian gene expression in cultured Rat-1 fibroblasts. Curr Biol 2000;10:1291-4.